

# A Wigner molecule at extremely low densities: a numerically exact study

Miguel Escobar Azor[1,2], Léa Brooke [1], Stefano Evangelisti [1*], Thierry Leininger [1], Pierre-François Loos [1], Nicolas Suaud [1] and J. A. Berger [1,3†]

**1** Laboratoire de Chimie et Physique Quantiques, IRSAMC, CNRS, Université de Toulouse, UPS, France
**2** Instituto de Ciencia Molecular, Universidad de Valencia, Spain
**3** European Theoretical Spectroscopy Facility (ETSF)

⋆ stefano.evangelisti@irsamc.ups-tlse.fr, † arjan.berger@irsamc.ups-tlse.fr

## Abstract

In this work we investigate Wigner localization at very low densities by means of the exact diagonalization of the Hamiltonian. This yields numerically exact results. In particular, we study a quasi-one-dimensional system of two electrons that are confined to a ring by three-dimensional gaussians placed along the ring perimeter. To characterize the Wigner localization we study several appropriate observables, namely the two-body reduced density matrix, the localization tensor and the particle-hole entropy. We show that the localization tensor is the most promising quantity to study Wigner localization since it accurately captures the transition from the delocalized to the localized state and it can be applied to systems of all sizes.

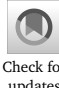

# 1  Introduction

When studying a system constituted of only electrons, one can consider two limiting regimes, that of high and low density. Since the kinetic energy is proportional to $r_s^{-2}$ while the electronic repulsion is proportional to $r_s^{-1}$, with $r_s$ the Wigner-Seitz radius (i.e., half the average distance between two electrons) at high electron density (small $r_s$), the kinetic energy dominates the electronic repulsion, and we have a Fermi gas. Instead, at low electron density (large $r_s$) the electronic repulsion dominates the kinetic energy and the electrons localize. It was predicted by Wigner [1] that, at very low density, the electrons crystallize on lattice sites. This is what is called a Wigner crystal. Experimental evidence has been reported for a liquid-to-crystal electron phase transition in a two-dimensional sheet of electrons on the surface of liquid helium. [2] In general, whenever, at low density, localization is due to the electronic repulsion we speak of Wigner localization. When only few electrons are involved, systems exhibiting Wigner localization are often referred to as Wigner molecules. [3–7] Recently, Wigner localization was observed experimentally in a 2-electron Wigner molecule. [8]

It is important to be able to describe Wigner localization within *ab initio* theory in order to analyze and perhaps even predict their properties. However, in these low-density regions the electron correlation is strong which is a problem for many condensed-matter theories. For example, Kohn-Sham density functional theory (KS-DFT) [9, 10] using currently available functionals fails to describe strong correlation satisfactorily. As an alternative to KS-DFT, the strictly correlated electrons (SCE) DFT has been proposed to deal with the strongly correlated limit. [11, 12] The SCE-DFT can be combined with KS-DFT and Wigner localization has been observed with this approach. [13–15] Instead, in this work we use the exact diagonalization of the many-body Hamiltonian to obtain numerically exact results. This allows us to explore the regime of extremely low densities, down to densities of the order of 1 electron per 10 $\mu m$.

In a recent work, we studied Wigner localization in a quasi-one-dimensional system in which the electrons were confined to a line segment with a positive background by placing three-dimensional gaussians along the line. [7, 16] We observed Wigner localization at low densities. Although the localization we observed was clearly due to the electronic repulsion, quantitatively the results were influenced by border effects. Moreover, we used a multi-purpose software for the numerical calculations due to which we were limited to system sizes smaller than 100 Bohr. Due to these restrictions we were not able to fully appreciate the delta-peak nature of the Wigner localization in the many-body wave function.

The goal of this work is therefore two-fold: i) We remove any border effects by confining the electrons to a ring [17–20] by placing the three-dimensional gaussians along the ring perimeter, and ii) We wrote a computer code that is specifically dedicated to treat low densities which allows us to reach ring perimeters as large as $10^6$ Bohr ($\approx 0.05\ mm$). We note that by confining the electrons to a ring also removes the need to add a positive background. [18] As described in the following, with this approach we are able to see the delta-peak nature of the localization in the many-body wave function.

Here we will limit our study to two electrons, since it is sufficient to observe the Wigner

localization while at the same time it allows us to obtain numerically exact results by exact diagonalization of the Hamiltonian. We will analyze several possible indicators of localization. In particular, we will study the 2-body reduced density matrix, the localization tensor and the particle-hole entropy. We use Hartree atomic units ($\hbar = e = m_e = 4\pi\epsilon_0 = 1$) througout.

## 2 Theory

### 2.1 The Hamiltonian

As mentioned in the Introduction we will study two electrons that are confined to a ring. The two electrons are confined by a potential that is implicitly defined by the basis set which consists of $M$ identical 3D $s$-type gaussians that are evenly distributed along the ring. These normalized gaussian functions are given by

$$g(\mathbf{r} - \mathbf{A}) = (2\alpha/\pi)^{3/2} \exp\left[-\alpha |\mathbf{r} - \mathbf{A}|^2\right], \tag{1}$$

in which $\mathbf{A}$ denotes the center of the gaussian and $\alpha$ its exponent which is directly linked to the width of the gaussian function ($\sim 1/\sqrt{\alpha}$). Since the gaussians are 3D also the ring is 3D, or rather, quasi-1D, since the width of the ring is much smaller than its perimeter. We, therefore, have the following Hamiltonian,

$$\hat{H} = -\frac{1}{2}\nabla_1^2 - \frac{1}{2}\nabla_2^2 + \frac{1}{r_{12}}, \tag{2}$$

in which the first two terms on the right-hand side are the 3D kinetic energy operators for electron 1 and 2, respectively. The last term is the repulsive 3D Coulomb potential in which $r_{12} = |\mathbf{r}_1 - \mathbf{r}_2|$ is the distance between the two electrons, i.e., the electrons interact through the ring.

We note the system with electrons confined to a ring with a perimeter of length $L$ might also be represented by electrons confined to a line of length $L$ and applying periodic boundary conditions. In that case the position operators in $r_{12}$ should be replaced by a complex position operator $q_L(x)$ that we have recently proposed. In 1D it is given by [21]

$$q_L(x) = \frac{L}{2\pi i}\left[\exp\left(\frac{2\pi i}{L}x\right) - 1\right]. \tag{3}$$

We represent the Hamiltonian in Eq. (2) in the basis of the evenly distributed gaussians which we then diagonalize to find the exact wave functions and eigenenergies. From the exact wave functions we can then obtain several exact observables of interest. We will mainly focus on the ground state of two electrons on the ring which is a spin singlet.

We have verified our approach and its implementation by comparing to analytical results which are available for one electron confined to a strictly 1D ring. They are given by

$$E_n^{\text{exact}}(R) = \frac{n^2}{2R^2}, \tag{4}$$

where $n$ is an integer and $R$ is the radius of the ring. When the width of the gaussians ($\sim 1/\sqrt{\alpha}$) is much smaller than $R$, the energy spectrum of the system tends to the energies in Eq. (4). We note, however, that some caution must be used when evaluating distances in the system, even in the limit $1/\sqrt{\alpha} \to 0$. In particular, when computing overlaps and kinetic energies, it is only when each distance is measured along an arc of the ring that in the limit $1/\sqrt{\alpha} \to 0$ the energies tend to those given in Eq. (4).

Since the density, by definition, has the same symmetry as the Hamiltonian it will have rotational symmetry. Therefore, the one-body density will be a constant as a function of the position on the ring, and unable to characterize the Wigner localization. However, for 2 electrons, the Wigner localization can be studied using the two-body density matrix, which shows the correlation between the positions of two electrons.

We note that the present scenario defines a floating Wigner crystal. Another scenario corresponds to a pinned Wigner crystal where the spatial symmetry of the wave function is broken by a small external perturbation or impurity. [22] In this case, the Wigner crystallization can be observed via the 1-body density. We will not consider such situations in the present manuscript.

### 2.2 The 2-body reduced density matrix

The $N$-body reduced density matrix ($N$-RDM) gives the conditional probability of the presence of $N$ electrons in space. The 2-body reduced density matrix $\Gamma^{(2)}$ is defined in 1D as

$$\Gamma^{(2)}(x_1, x_2; y_1, y_2) = \frac{(N-1)N}{2} \int dx_3 \cdots dx_N \Psi^*(y_1, y_2, x_3, \cdots, x_N) \Psi(x_1, \cdots, x_N), \quad (5)$$

in which $\Psi$ is an $N$-body wave function. In particular, its diagonal elements $\Gamma(x_1, x_2) = \Gamma^{(2)}(x_1, x_2; x_1, x_2)$ give the probability of having an electron at $x_1$ if a second electron is located at $x_2$. For two electrons it is given by

$$\Gamma(x_1, x_2) = |\Psi(x_1, x_2)|^2. \quad (6)$$

In the present quasi-1D context, the 2-RDM plays a crucial role in measuring the locality of the electrons. Indeed, because of the rotational invariance of the wave function the 1-RDM is a constant, regardless of the nature of the wave function, since all points in space are equivalent. It is the 2-RDM, on the other hand, that is able to indicate if the electrons are strongly correlated (Wigner localization), or weakly correlated (Fermi gas).

The situation is rather different, on the other hand, for 2D and 3D generalizations of electrons on a ring (1-torus), i.e., electrons on a 2-torus and a 3-torus. [1] Just as the diagonal of the 1-RDM is a constant for a ring due to symmetry, the diagonal of the 2-RDM is a constant for the 2-torus. This implies that the 2-RDM is unable to give a complete characterization of a regular crystalline structure on the 2-torus. Similarly, the 3-RDM is a constant for the 3-torus, and is unable to characterize the crystallisation of the electrons. Higher-order density matrices would be in principle needed for this purpose, e.g. the 3-RDM for 2-tori, and the 4-RDM for 3-tori, etc. However, the evaluation of higher-order density matrices are computationally extremely demanding, and their calculation would become unfeasible even for very small systems. In such a situation, other indicators, like the electron entropy, and in particular the localization tensor, would be much more practical.

### 2.3 The localization tensor

The localization tensor distinguishes between metallic and insulating behavior. It was developed by Resta and co-workers [23–26] (see also Ref. [27]) and is based on an idea of Kohn [28] to describe the insulating state from the knowledge of the ground-state wave function (see also Ref. [29]). The localization tensor has been applied to study the metallic behavior of clusters [30–38], chemical bonding [39] and electron transport. [40] It has recently also been used to investigate Wigner localization. [7]

---

[1]A $D$-torus, the $D$-dimensional version of the torus, is the product of $D$ circles.

The localization tensor $\boldsymbol{\lambda}$ is defined as the total position spread tensor $\boldsymbol{\Lambda}$ normalized with respect to the number of electrons $N$, i.e.,

$$\boldsymbol{\lambda} = \frac{\boldsymbol{\Lambda}}{N}. \tag{7}$$

The total position spread tensor is defined as the second moment cumulant of the total position operator,

$$\boldsymbol{\Lambda} = \langle\Psi|\hat{\mathbf{R}}^2|\Psi\rangle - \langle\Psi|\hat{\mathbf{R}}|\Psi\rangle^2, \tag{8}$$

in which the total position operator $\hat{\mathbf{R}}$ is defined by

$$\hat{\mathbf{R}} = \sum_{i=1}^{N} \hat{\mathbf{r}}_i, \tag{9}$$

where $\hat{\mathbf{r}}_i = \mathbf{r}_i$ the standard position operator for electron $i$.

The three diagonal elements of $\boldsymbol{\Lambda}$ are the variances of the wave function in the $x$, $y$, and $z$ directions. Therefore, these elements are large when the electrons are delocalized and small when they are localized. This shows that the localization tensor is able to distinguish between a conducting and an insulating behavior. Moreover, in the thermodynamic limit the localization tensor diverges for conductors while it remains finite in the case of insulators. We note that the second term on the right-hand side of Eq. (8) ensures gauge invariance with respect to the choice of the origin of the coordinate system.

Due to the symmetry of the ring we have $\lambda_{xx} = \lambda_{yy}$ and $\lambda_{zz} \ll \lambda_{xx}$. In the following we will focus on the trace of $\boldsymbol{\lambda}$, i.e.,

$$\lambda = \mathrm{Tr}\{\boldsymbol{\lambda}\}. \tag{10}$$

## 2.4 The particle-hole entropy

The fractionality of the natural occupation numbers, i.e., the eigenvalues associated with the 1-RDM, can be related to the amount of electron correlation in a system. [41, 42] Therefore, the particle-hole entropy has been proposed as a measure of the presence of correlation in a system. [43, 44] In the case of a pure state described by a wavefunction $\Psi$, the particle-hole entropy is defined as:

$$S = \sum_{j=1}^{M} \Big[ \underbrace{-n_j \ln n_j}_{S_{\text{part}}} \underbrace{-(1-n_j)\ln(1-n_j)}_{S_{\text{hole}}} \Big], \tag{11}$$

where the sum runs over the $M$ natural spinorbitals of $\Psi$, and $n_j$ is the occupation number of spinorbital $\phi_j$. The natural spinorbitals are the eigenfunctions of the one-body reduced spin-density matrix and the occupations numbers are its eigenvalues. The first and second terms in the summation are the particle and hole contributions, $S_{\text{part}}$ and $S_{\text{hole}}$, respectively, to the total entropy. While the entropy of a single determinant is zero, since all the occupation numbers are either 0 or 1, the entropy has its maximum value when all the spinorbitals have equal occupation numbers. Therefore, the particle-hole entropy of a Fermi gas will be small while it will be large in the regime of Wigner localization. In particular, this will be the case for an $S_z = 0$ wave function because of the large number of Slater determinants that contribute to the wave function with similar weight. Indeed, we have an $S_z = 0$ ground-state wave function.

The dependence of the particle-hole entropy on the natural spinorbitals implies a dependence on the basis set. For large densities this dependence is negligible but in the limit of Wigner localization there is a strong dependence on the basis set. This dependence can be made explicit as we will now show. All the occupation numbers become identical when the

length of the perimeter of the ring tends to infinity and the electrons localize. This leads to an upper bound for the entropy in the limit $L \to \infty$. Let us consider $N$ electrons in $M$ spinorbitals. In the limit $L \to \infty$ each spinorbital $\phi_j$ has occupation number $n_j = N/M$. Therefore, we obtain the following upper bounds for the particle and hole entropies in the limit $L \to \infty$,

$$\lim_{L\to\infty} S_{\text{part}} = -N \ln(N/M), \tag{12}$$

$$\lim_{L\to\infty} S_{\text{hole}} = -(M-N)\ln(1-N/M). \tag{13}$$

If we subsequently let the number of gaussians, and thereby the number of spinorbitals $M$, tend to infinity, we see that particle entropy diverges logarithmically as

$$\lim_{L\to\infty} S_{\text{part}} \sim -N \ln(N/M). \tag{14}$$

Instead, in this limit, the hole entropy tends to a constant, namely the number of electrons,

$$\lim_{M\to\infty} \lim_{L\to\infty} S_{\text{hole}} = N. \tag{15}$$

The total entropy therefore behaves as

$$S = -N \ln(N/M) + N + \mathcal{O}(M^{-1}) \tag{16}$$

for large $L$ and $M$.

## 3 Computational details

As mentioned before, we study a quasi-1D periodic system of electrons by placing a series of identical 3D gaussian functions [see. Eq. (1)] to form a ring. The centers of the gaussians are equally spaced. It can be shown that the overlap between two neighboring gaussians is proportional to the parameter $\xi = \alpha\delta^2$, where $\alpha$ is the exponent of the gaussian and $\delta$ is the distance along the arc between two neighboring gaussians. [16] The overlap should be sufficiently large to be able to accurately describe the electronic wave function but not too large to avoid numerical problems due to a quasi-linearly dependent basis functions. We have demonstrated that a value of $\xi \approx 1$ is optimal for quasi-1D systems. [16] In this work we used $\xi = 1$. We have used 128 equidistant gaussians which was sufficient to obtain converged results for rings with lengths up to $10^6$ Bohr.

For two electrons the spin wave function can correspond either to a single or a triplet. Therefore, it is convenient to generate spin-adapted wave functions. This is particularly important at low density, where the singlet and triplet wave functions tend to become degenerate. In absence of spin adaptation, due to numerical errors, the diagonalization procedure could mix the two quasi-degenerate states, and therefore the computed properties would correspond to wave functions that are not eigenfunctions of $\hat{S}^2$. The details of the diagonalization of the Hamiltonian can be found in the appendix. There we also discuss how one can take advantage of the rotational symmetry of the problem to reduce the numerical cost of the calculation.

## 4 Results

We now report the results for 2 electrons confined to a quasi-1D-ring.

## 4.1 The 2-body reduced density matrix

As mentioned earlier, for a system without translational and/or rotational symmetry, such as a linear system within open-boundary conditions, a symmetry-broken system or a system with an impurity (i.e. pinned electrons), the electron density is sufficient to characterize the Wigner localization. [7] However, as explained in the previous section, in the case of a ring, due to the rotational symmetry of the Hamiltonian, the density is no longer a good observable to identify the Wigner localization. Instead, we can use the 2-body reduced density matrix to characterize this localization since it describes the correlation between the positions of two electrons.

In Fig. 1, we report the two-body reduced density matrix $\Gamma(0, x)$ as a function of $x$ for different values of the length of the perimeter $L$ of the ring. It gives the probability amplitude of finding an electron at $x$ while another electron is present at $x = 0$. [2]

At large electron density (small $L$), $\Gamma(0, x)$ is almost constant, since the kinetic energy dominates the electronic repulsion. This is the Fermi-gas regime. Instead, at low electron density (large $L$), the second electron has the largest amplitude at $x = L/2$, i.e., exactly at the position that is opposite to the position of the first electron. The electronic repulsion is dominant and pushes the two electrons to opposite positions on the ring. By increasing $L$ the two-body reduced density matrix becomes ever more localized. For very large values of $L$ the density matrix becomes close to a delta function. There is the formation of a 1D electron "lattice", we observe the Wigner localization. It is difficult to pinpoint exactly for which length the transition from the Fermi to the Wigner regime occurs. Nevertheless, the Wigner localization becomes apparent for lengths of the order of $L = 10$ Bohr.

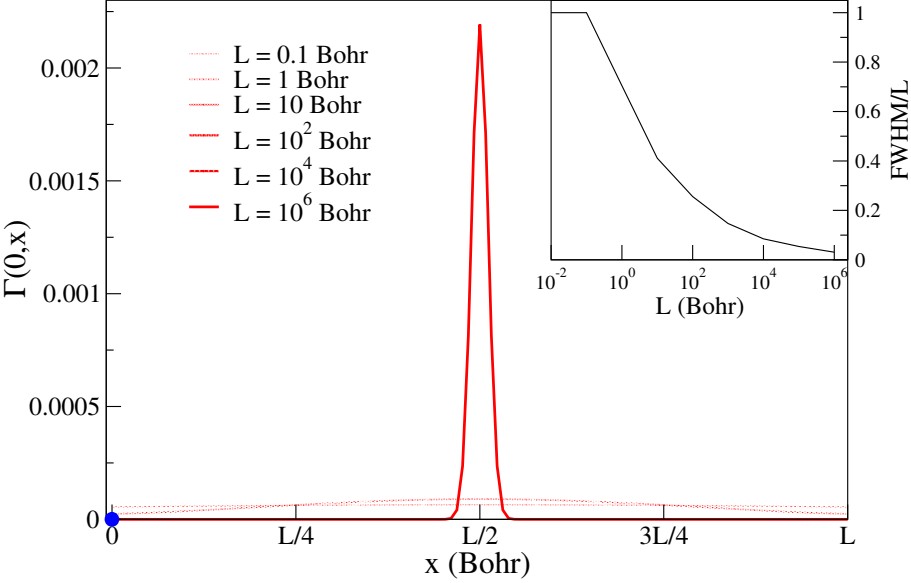

Figure 1: The two-body reduced density matrix $\Gamma(0, x)$ for two electrons on a ring for various values of the length $L$ of the perimeter. The position of the first electron is fixed at $x = 0$ (indicated by the blue dot). Inset: Full-width at half maximum (FWHM) of $\Gamma(0, x)$ normalized with respect to $L$ as a function of $L$. For small $L$ the FWHM is not well-defined and the normalized FWHM is set to 1.

However, as mentioned before, the two-body reduced density is not sufficient to characterize Wigner localization for systems of higher dimensions. Therefore, we will study two other

---

[2]To be precise, the results in Fig. 1 are the 2-RDM expressed in the basis of the orthonormal gaussians, i.e., $\Gamma_{i\alpha j\beta i\alpha j\beta} = \langle \Psi | a_{i\alpha}^\dagger a_{j\beta}^\dagger a_{i\alpha} a_{j\beta} | \Psi \rangle$ in which the first electron is fixed at $i = 1$.

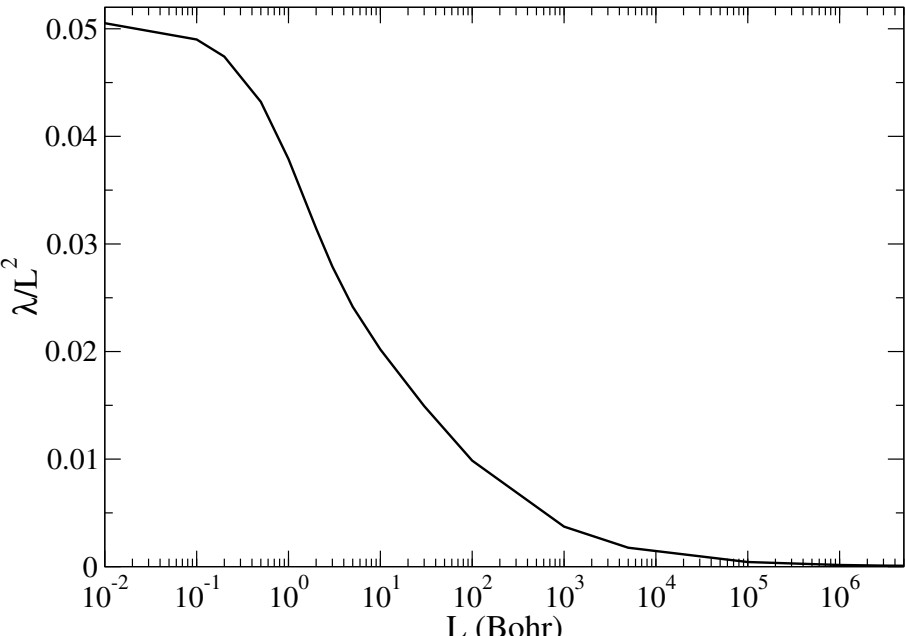

Figure 2: The trace of the localization tensor $\lambda/L^2$ as a function of the length $L$ of the perimeter of the ring.

indicators of the Wigner localization, namely the localization tensor and the electron-hole entropy. The advantage of these quantities is that they can also be easily calculated for systems of higher dimensions and with many electrons.

## 4.2 The localization tensor

In order to compare localization tensors for systems of different sizes, we report in Fig. 2 the dimensionless quantity $\lambda/L^2$ as a function of the length $L$ of the perimeter of the ring.

We see that for large density ($L < 0.1$) $\lambda/L^2$ is almost constant while its value starts to decrease for $L > 1$ Bohr. This marks the beginning of the transition to a localized state. For very low density, $L \gg 1$, the localization tensor almost vanishes, clearly indicating the Wigner localization. The behavior of the localization tensor is in agreement with the 2-body reduced density matrices reported in Fig. 1, i.e, the transition from the Fermi-gas regime to the Wigner regime occurs in the region around $L = 10$ Bohr. Finally, we note that the qualitative behavior of the localization tensor we observe here for the ring is very similar to the behavior we observed for a linear system within open boundary conditions. [7] Hence, the localization tensor seems to be a robust indicator of the transition to the Wigner regime.

## 4.3 The particle-hole entropy

In Fig. 3, we report the particle-hole entropy as a function of the length $L$ of the perimeter of the ring. For large average densities ($L \ll 1$) the entropy $S$ is very small since the Fermi gas can be accurately described by a single Slater determinant. The entropy starts to rapidly increase for $L > 1$ Bohr, indicating the transition towards the Wigner regime.

Up to about $L = 10$ Bohr the entropy is qualitatively similar to the behavior of the entropy for a linear system. [7] However, while for the linear system the entropy stabilizes for $L > 10$ Bohr, for the ring it continues to increase logarithmically. This is due to the particle part of the entropy which grows logarithmically in the region $10^2 < L < 10^6$ while the hole part saturates

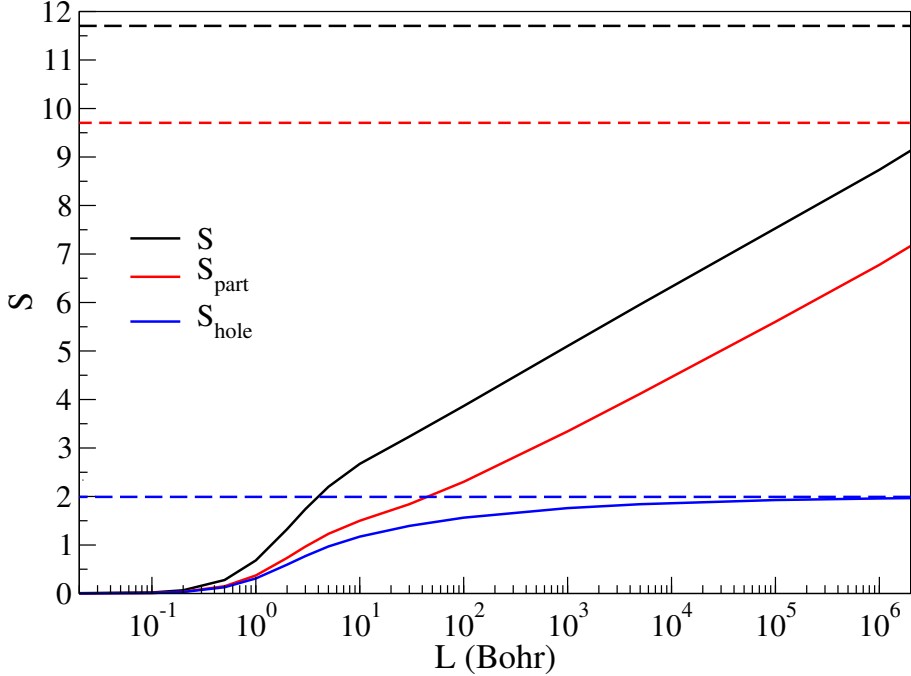

Figure 3: The entropy $S$ as a function of the length $L$ of the perimeter of the ring. The dashed lines indicate the asymptotic limits of the entropies for 128 gaussians when $L \to \infty$ [see Eqs. (12) and (13)].

in this region.

Although it might appear that the entropy diverges logarithmically with $L$, this is not the case. Using Eqs. (12) and (13) we can determine the asymptotic limits of the entropy and its two contributions (for a given $M$). They are finite and we reported them in Fig. 3. In order to reach the asymptotic limit of the total entropy we have to go beyond $L = 10^6$ Bohr which would require a larger number of gaussians to guarantee stable results. However, a larger number of gaussians yields a larger number of spinorbitals which will raise the asymptotic limit, requiring an even larger number of gaussians to reach it, etc. Due to this vicious circle, the particle-hole entropy is not capable of capturing the localized state. Therefore, the particle-hole entropy seems less convenient than the localization tensor as an indicator of Wigner localization.

## 5 Conclusion

We have investigated Wigner localization at extremely low densies using an exact diagonalization of the many-body Hamiltonian for a system of two electrons confined to a ring. Due to the rotational symmetry of the system, Wigner localization cannot be observed in the local density. Therefore, we have studied alternative quantities, namely, the two-body reduced density matrix, the localization tensor and the particle-hole entropy. We have clearly observed the Wigner localization both in the two-body reduced density matrix and in the localization tensor. Instead, in the particle-hole entropy the Wigner localization cannot easily be detected. With respect to the two-body reduced density matrix, the advantage of the localization tensor is that it can also be applied without increased difficulty to systems with more dimensions and

more electrons.

This work paves the way for further investigations of Wigner localization. In particular, it would be interesting to study two-dimensional electron systems with more than two electrons, going towards a true many-electron Wigner crystal that can be experimentally observed. However, to make these calculations numerically feasible, we have to make use of the translational symmetry, the sparsity of the two-electron integrals, etc. Work along this direction is in progress. We note that in order to exploit the translational symmetry one can impose periodic boundary conditions (PBC). Unfortunately, the standard position operator, which appears in the localization tensor, is not compatible with PBC. However, we have recently proposed an alternative position operator that is compatible with PBC. [21] This would allow us to pinpoint the phase transition which is well-defined only in the thermodynamic limit. Finally, it could be of interest to combine our approach with the density-matrix renormalization group (DMRG) algorithm to treat 1D systems with many electrons [45, 46] and to generalise our method to finite temperature [47–49]

# Acknowledgments

This study has been supported through the EUR grant NanoX ANR-17-EURE-0009 in the framework of the "Programme des Investissements d'Avenir"

The authors dedicate this article to Jean-Paul Malrieu in honor of his 80th birthday for the considerable advances in theoretical chemistry that his work allows.

# A   Diagonalization of the Hamiltonian

We describe here the details of the diagonalization procedure that yields the eigenvalue and eigenvectors of the Hamiltonian matrix. We wrote two versions of the program: i) a version based on the local gaussian functions and ii) a version that takes advantage of the rotational symmetry of the system. Both versions yield identical results.

## A.1   Local gaussian functions

When using the local gaussian functions, the number of determinants having the spin projection $S_z = 0$ is given by $n^2$, where $n = M/2$ is the number of gaussian functions. The orbitals are orthogonalized with a $S^{-1/2}$ symmetrical procedure and then the Hamiltonian matrix is built by using the Slater-Condon rules. [50] As mentioned in the main text, for two electrons the spin wave function can correspond either to a single or a triplet. Therefore, it is convenient to generate spin-adapted wave functions. This is particularly important at low density, where the singlet and triplet wave functions tend to become degenerate. In absence of spin adaptation, due to numerical errors, the diagonalization procedure could mix the two quasi-degenerate states, and therefore the computed properties would correspond to symmetry-broken states. The spin-adaptation procedure is straightforward, since it corresponds to taking the symmetric (singlets) and anti-symmetric (triplets) combinations of the Slater determinants $|g_{i,\alpha} g_{j,\beta}\rangle$ built with the gaussians $g_i$:

$$|\Phi_{ij}^{[S]}\rangle = |g_{i,\alpha} g_{j,\beta}\rangle + |g_{i,\alpha} g_{j,\beta}\rangle, \tag{17}$$

$$|\Phi_{ij}^{[T]}\rangle = |g_{i,\alpha} g_{j,\beta}\rangle - |g_{i,\alpha} g_{j,\beta}\rangle, \tag{18}$$

for singlets and triplets, respectively. We note that for triplets we must have $i \neq j$. The Hamiltonian matrix reduces therefore to two spin adapted diagonal blocks, of dimensions

$n(n+1)/2$ and $n(n-1)/2$, respectively (note that $n(n+1)/2 + n(n-1)/2 = n^2$).

## A.2 Rotational symmetry

When exploiting the rotational symmetry $C_n$ of the problem the wave function has a particularly simple structure. Symmetry-adapted orbitals $\phi_k$ are built as linear combinations of the gaussians. Since all the (orthogonalized) gaussians are equivalent under a rotational symmetry operation, each symmetry-adapted orbital is associated to a different quasi-momentum $k = 2\pi\kappa/L$, with $\kappa$ an integer going from 0 to $n-1$, where $n$ is the number of unit cells. The quasi-momenta being additive, we have that the combination of an $\alpha$-electron in spinorbital $|\phi_{k,\alpha}\rangle$ with a $\beta$-electron in spinorbital $|\phi_{k',\beta}\rangle$ gives rise to a Slater determinant having total quasi-momentum $K = k + k'$. Since the Hamiltonian is totally symmetric, only determinants having the same total quasi-momentum $K$ yield nonzero expectation values of the Hamiltonian. Therefore, for a ground state with total rotational symmetry $K$, the wave function will be a linear combination of determinants, $|\phi_{k+K,\alpha}\phi_{-k,\beta}\rangle$, i.e.,

$$|\Psi_K\rangle = \sum_k C_{k+K,-k}|\phi_{k+K,\alpha}\phi_{-k,\beta}\rangle. \tag{19}$$

The important point here is that the number of coefficients $C_{k+K,-k}$ is identical to the number of orbitals, and not to its square. Therefore, the coefficients can be obtained by diagonalizing matrices (one matrix for each $K$) of order $n$, with a huge computational gain. Because of the symmetry properties of the singlet and triplet wave functions, we have

$$C_{k+K,-k} = C_{-k+K,k}, \tag{20}$$
$$C_{k+K,-k} = -C_{-k+K,k}, \tag{21}$$

for singlets and triplets, respectively.

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
