# Peer review of "A Wigner molecule at extremely low densities: a numerically exact study"

_SciPost Physics, doi:SciPost Phys. Core 1, 001 (2019)_

## Round 1 · Referee Report · Anonymous (Referee 1) · 2019-7-9

Report

The paper presents a study of the electron localization as a function of the electron density, focusing in particular on the Wigner crystallization (WC), for a quasi-1D system composed of two electrons Gaussian-confined on a ring of perimeter L. Full understanding and characterization of the Wigner crystallization is an open problem because of the difficulty in the description of correlations in the framework of the most general unsolved quantum many-body problem. The present work is along this research line, and therefore interesting per se.

The work is scientifically sound and the methodology valid. It is, to the best of my knowledge, original. To the extent I could check, I did not find any issue, mistake, or error that can, bona aut mala fide, affect the findings. I anyway warn the authors that I do not assume any responsibility for possible errors which therefore fall under their full liability.
I solicit the editor to publish the paper.

In the following minor remarks/comments/suggestions which might be correct or wrong and that the authors may optionally take into account, or discard at all if they will find them fully irrelevant:

I personally found the work really interesting (per se, as I said above) and stimulating further developments along the same line. For this reason, I don't understand the authors' need to add an introduction bringing on a "technological interest" of the present study, making a link with "quantum dots" and their "tuning" to make them "promising candidates for qubits", up to even evoking the legendary and mythological "quantum computer". These introductions are in the style required by editors of those high-IF journals with divulging character. But on high-level journals, like SciPost, whose readership is composed by highly competent scientists who can perfectly understand the importance of the present work, such introductions are an offense to their intelligence. I think that they are unworthy and reduce the scientificity of the article.
It is not immediately clear from the title, the abstract and even the introduction which is the system studied precisely. Indeed, until section II.A the reader is uncertain if the system is a really 1D ring with 1D kinetic and interaction terms. Or for quasi-1D the authors mean a 3D system but rigidly confined on a ring (in analogy with spherium with electrons rigidly confined on a 2D manifold but with a 3D interaction). I think that a more immediately comprehensible definition for the present system is "a quasi-1D system composed of two electrons Gaussian-confined on a ring of perimeter L". I think that this is an important information to be quickly delivered for a reader to promptly classify this work with respect to his interests.
I don't know if the use of the term "ab initio" in the title is appropriated in this case: ab initio normally refers to treatment of all (or a maximum relevant) of the degrees of freedom in a real system by referring to the microscopical (and not a model) Hamiltonian. The present is rather a model. Even Wigner, Nozieres et al., when studying the 3D electron-liquid jellium model, did not define it an ab initio study. And jellium has much more close realizations in nature (e.g. bulk sodium or lithium) than the present model. On the other hands, I agree that the present study "is important to be able to describe Wigner localization within ab initio theory".
The present work casts some interesting doubts on the ab initio quantum chemistry (full-CI) Gaussian methodology, that is whether it possesses a real full 3D ab initio valence or it should be rather considered a reduced quasi-1D study. As it emerges here, a Gaussian basis-set, with Gaussians appropriately placed along a ring, in practice makes the system quasi-1D. Then also what in quantum chemistry are defined as ab initio full-CI studies of molecules, for examples in dimers with Gaussians placed at the two atomic positions or at best placing a further midbond set, might be rather receiving a strong bias toward a quasi-1D situation. And so, for the use of the Gaussian basis-set itself, they deviate from a genuine description of the real 3D system. It would be interesting if the authors could provide the comparison of their numerical study as a function of Gaussian width $\alpha$, with the available analytical study on the rigid 1D ring: the case of one electron is already sufficient to show how much, depending on $\alpha$, a Gaussian study is closer to the quasi-1D or rather evolves towards the full 3D.
It is to me evident that the diagonal of the 1-RDM (the density) is constant, and only the diagonal of the 2-RDM can be used to characterize a WC, but it is not clear why raising the dimensionality of the system (from the 1-torus/ring to the 2 and 3-torus) one needs higher order RDMs (3-RDM and 4-RDM). I would have said that the 2-RDM is sufficient always. Is this valid also for the full 2D and 3D electron liquids not confined on the torus? Can the authors provide an intuitive explanation, or at least a reference where this can be understood?
Since the beginning I had the impression that the particle-hole entropy is a quantity that strongly depends on the number M of natural orbitals taken into account, and so strongly depends on the basis-set size. At least much more than the two other considered indicators. If the authors confirm this impression of mine, than since the beginning it is evident the difficulty in using this quantity as a robust indicator of WC.
Finally, from this work it does not emerge a net criterion to exactly pinpoint the density at which the phase transition to the Wigner crystal really occurs. If one looks to the 2-RDM, for small L the FWHM is not well defined, so that the normalized FWHM is manually set to 1: therefore, we cannot take the WC phase transition to occur when the normalized FWHM departs from 1. The trace of the localization tensor does not achieve a well defined plateau for high densities, and so what to take for phase-transition boundary? I have the impression that the system is a Wigner crystal even at high densities and since the beginning, so that there is no real phase transition to another phase. Isn't this really the case?
  • validity: -
  • significance: -
  • originality: -
  • clarity: top
  • formatting: perfect
  • grammar: -

Author:  Arjan Berger  on 2019-07-23  [id 571]

(in reply to Report 1 on 2019-07-09)

We thank the referee for the report and we are glad that the referee thinks our work is interesting.

Below we reply to the comments and questions of the referee.

The referee writes:

"I personally found the work really interesting (per se, as I said above) and stimulating further developments along the same line. For this reason, I don't understand the authors' need to add an introduction bringing on a "technological interest" of the present study, making a link with "quantum dots" and their "tuning" to make them "promising candidates for qubits", up to even evoking the legendary and mythological "quantum computer". These introductions are in the style required by editors of those high-IF journals with divulging character. But on high-level journals, like SciPost, whose readership is composed by highly competent scientists who can perfectly understand the importance of the present work, such introductions are an offense to their intelligence. I think that they are unworthy and reduce the scientificity of the article."

Authors' reply:

We agree with the referee that mentioning the potential technological interest of our work is not necessary. However, it could be useful for the reader to have an idea about potential applications of our approach. We think that quantum dots are within reach of our approach and we plan to study them in the future. However, we agree with the referee that the application to quantum computers could be considered farfetched. We will consider rewriting this part in the revised manuscript.

The referee writes:

"It is not immediately clear from the title, the abstract and even the introduction which is the system studied precisely. Indeed, until section II.A the reader is uncertain if the system is a really 1D ring with 1D kinetic and interaction terms. Or for quasi-1D the authors mean a 3D system but rigidly confined on a ring (in analogy with spherium with electrons rigidly confined on a 2D manifold but with a 3D interaction). I think that a more immediately comprehensible definition for the present system is "a quasi-1D system composed of two electrons Gaussian-confined on a ring of perimeter L". I think that this is an important information to be quickly delivered for a reader to promptly classify this work with respect to his interests."

Authors' reply:

We agree with the referee. In the revised manuscript we will make it clear from the outset, i.e., in the introduction, that we are dealing with a 3D system.

The referee writes:

"I don't know if the use of the term "ab initio" in the title is appropriated in this case: ab initio normally refers to treatment of all (or a maximum relevant) of the degrees of freedom in a real system by referring to the microscopical (and not a model) Hamiltonian. The present is rather a model. Even Wigner, Nozieres et al., when studying the 3D electron-liquid jellium model, did not define it an ab initio study. And jellium has much more close realizations in nature (e.g. bulk sodium or lithium) than the present model. On the other hands, I agree that the present study "is important to be able to describe Wigner localization within ab initio theory"."

Authors' reply:

We agree with the referee and we will remove ab initio from the title.

The referee writes:

"The present work casts some interesting doubts on the ab initio quantum chemistry (full-CI) Gaussian methodology, that is whether it possesses a real full 3D ab initio valence or it should be rather considered a reduced quasi-1D study. As it emerges here, a Gaussian basis-set, with Gaussians appropriately placed along a ring, in practice makes the system quasi-1D. Then also what in quantum chemistry are defined as ab initio full-CI studies of molecules, for examples in dimers with Gaussians placed at the two atomic positions or at best placing a further midbond set, might be rather receiving a strong bias toward a quasi-1D situation. And so, for the use of the Gaussian basis-set itself, they deviate from a genuine description of the real 3D system. It would be interesting if the authors could provide the comparison of their numerical study as a function of Gaussian width α, with the available analytical study on the rigid 1D ring: the case of one electron is already sufficient to show how much, depending on α, a Gaussian study is closer to the quasi-1D or rather evolves towards the full 3D."

Authors' reply:

Depending on the problem at hand, i.e., the system and physical property under study, one chooses the most convenient basis set. For atoms and (small) molecules the most convenient basis is, in general, an atom-centered Gaussian basis set. This indeed given a bias on the results, but, in general, we would expect it to be a very small bias. The problem discussed in this work is slightly different in this respect, since there are no nuclei. The positions of the centers of the Gaussians define the system, i.e., a ring.
The test that the referee proposes is interesting but difficult to do in practice since evolving “towards the full 3D” would mean increasing the width of the Gaussians by reducing α, thereby increasing the overlap between the Gaussians, which will lead to numerical problems due to quasi-linear dependence.

The referee writes:

"It is to me evident that the diagonal of the 1-RDM (the density) is constant, and only the diagonal of the 2-RDM can be used to characterize a WC, but it is not clear why raising the dimensionality of the system (from the 1-torus/ring to the 2 and 3-torus) one needs higher order RDMs (3-RDM and 4-RDM). I would have said that the 2-RDM is sufficient always. Is this valid also for the full 2D and 3D electron liquids not confined on the torus? Can the authors provide an intuitive explanation, or at least a reference where this can be understood?"

To explain this point we ask the referee to consider, as an example, the hexagonal 2D Wigner crystal. If we fix the position of an electron E in the 2-RDM, the nearest-neighbor electrons will have the highest probability density when they are positioned on the vertices of a hexagon with E at its center. However, there an infinite number of hexagons with E at its center, all having equal probability. Therefore, the 2-RDM corresponding to an electron in E will be given by a set of circles, all having E as center, and each one having a constant value of the 2-RDM. For this reason, we need the 3-RDM, because fixing the position of a second electron, for example at one of the vertices of a hexagon around E, will fix the positions of the other electrons in that hexagon (and beyond). With a similar argument, we see that the 4-RDM is needed to characterize a 3D crystal.

The referee writes:

"Since the beginning I had the impression that the particle-hole entropy is a quantity that strongly depends on the number M of natural orbitals taken into account, and so strongly depends on the basis-set size. At least much more than the two other considered indicators. If the authors confirm this impression of mine, than since the beginning it is evident the difficulty in using this quantity as a robust indicator of WC."

Authors' reply:

We agree with the referee that the particle-hole entropy strongly depends on the basis-set size and that, therefore, one could expect it to be, a priori, not the most robust indicator to characterize Wigner localization. However, in a linear quasi-1D system we studied in a previous work [J. Chem. Phys.148, 124103 (2018)] it turned out to be a useful quantity to characterize the Wigner localization. Therefore, we included the study of the particle-hole entropy also in this work. In the revised manuscript we will add a sentence in section II.D to stress the dependence of the particle-hole entropy on the basis set and the implication on its robustness.

The referee writes:

"Finally, from this work it does not emerge a net criterion to exactly pinpoint the density at which the phase transition to the Wigner crystal really occurs. If one looks to the 2-RDM, for small L the FWHM is not well defined, so that the normalized FWHM is manually set to 1: therefore, we cannot take the WC phase transition to occur when the normalized FWHM departs from 1. The trace of the localization tensor does not achieve a well defined plateau for high densities, and so what to take for phase-transition boundary? I have the impression that the system is a Wigner crystal even at high densities and since the beginning, so that there is no real phase transition to another phase. Isn't this really the case?"

Authors' reply:

At high densities the system is a Fermi gas while it becomes a Wigner crystal at low densities. This can be seen from the diagonal of the 2-RDM in Fig. 1. It is close to a constant at high density, indicating that the two electrons behave independently while it is nonzero only around L/2 (the first electron is fixed at 0) at low density indicating that the two electrons are strongly correlated.
However, we should probably speak of a Fermi-gas-like and a Wigner-crystal-like system, since Fermi gases and Wigner crystals pertain to many-electrons systems i.e., systems with ~10^23 electrons. For the same reason it is difficult to pinpoint the phase transition which is only well-defined in the thermodynamic limit. Such a system is best represented within periodic boundary conditions. This is out of the scope of this work but it is something we will investigate in the future.

---

## Round 1 · Referee Report · Javad Vahedi (Referee 2) · 2019-8-5

Report

Comments to the Author

The authors study by means of exact diagonalization method the Wigner localization at very low density of a quasi one dimensional ring. Three observables, two-body reduced density matrix, localisation tensor, and particle-hole entropy are used to capture the Wigner localisation transition. The authors conclude with introducing the localisation tensor as more accurate and versatile, mainly for higher dimension, observable than the others.

The methods used in this study are well established and the results
physically sound. Although the present work does not extend much the previous work’s of authors (ref. 17 in the manuscript), but I believe the present article has a broad appeal for the community of researchers working in this field due to its pedagogical character and clarity. Thus I think that the paper can be accepted for publication once the authors address the following few points:

Requested changes

1) I would suggest to add a section or appendix about the details of numerical, for the pedagogical purpose.

2) I think that the authors should discuss more about transition point!!!, whether it can be found with the introduced observables or not?

---

## Round 1 · Referee Report · Anonymous (Referee 3) · 2019-8-12

Weaknesses

This is actually a pretty simple work, following up on a previous investigation [17] by the same group.

Report

The authors study the behavior of two electrons that are implicitly confined to a ring of varying diameter by the choice of computational basis. The main differences with respect to the previous work Ref. [17] from the same group are: * Ring geometry instead of open line segment. * No positive background charge. * Refined spatial resolution thanks to a customized code. * 2 instead of up to 8 electrons. The authors claim that this study allows them to detect signatures of a Wigner crystallization. However, for a true Wigner crystal, one would need a many-electron system while the present work considers only the relatively simple two-particle problem. Given that this is a follow-up study to Ref. [17] and that the problem investigated is a simple two-particle problem, I am not sure if this manuscript contains sufficient substance for publication in SciPost Physics. In any case, the authors should choose a title that makes the two-particle problem explicit. The terminology "Wigner molecule" (see end of the second paragraph of the Introduction) might be an option.

Apart from this, chapter III is a bit short, in particular for a work whose main merit is supposed to be a specially written dedicated code. Specifically, I would find the following information useful: * Was the rotational symmetry of the ring exploited? In other words: conservation of angular momentum should allow to reduce the two-electron problem to an effective one-body problem. * What is the structure of the matrix representation of the Hamiltonian Eq. (2) represented in the basis Eq. (1)? Specifically: is it a sparse matrix or can it be approximated by a sparse matrix to numerical accuracy? * What algorithm was used to find the ground state? A library routine or an iterative algorithm?

Further questions/comments: * The references [17,38-46] on applications of the "localization tensor" are all from the group around Stefano Evangelisti. If this tool was also used by other groups, appropriate references should be added. * I did not find a definition of "natural spinorbitals" such that the particle-hole entropy Eq. (11) is ill defined. A related comment concerns the particle part of the entropy in Fig. 3. It seems to me that the asymptotic value (2ln(128) ?) measures just the number of basis functions and thus is not intrinsic to the problem. If so, the asymptotic value shown in Fig. 3 is actually a bit misleading. * The units of the full-width at half maximum in Fig. 1 are not clear (I suspect that it is normalized by the length L). * It is good to have a perspective on two dimensions in the Conclusions. Among others, this justifies some of the more general discussions throughout the manuscript. However, it would also be good to have the authors' opinion on the possibility to go beyond two particles to the true many-body Wigner crystal.

Requested changes

1) Adapt title to two-particle problem studied (e.g., using "Wigner molecule"). 2) Expand section III "Computational details". 3) Add references on applications of the "localization tensor" by other authors (?). 4) Define "natural spinorbitals". 5) Specify units of the full-width at half maximum in Fig. 1. 6) Clarify impact of number of basis functions used on the asymptotic value of the particle entropy in Fig. 3. 7) Add comments on the possibility to go beyond two particles to the true many-body Wigner crystal to the Conclusion.

---

## Round 2 · Referee Report · Anonymous (Referee 1) · 2019-9-11

Report

I don't understand why the editor-in-charge asked me to review the manuscript again when in my previous report I "solicited" the journal to publish the paper, so that the whole community could benefit, without delay, of the better comprehension brought by the present work. However, I had a look at the other referees' reports and saw that the request was probably due to the fact that referee 3 has not agreed on publication, claiming it is a "follow-up" to Ref. [17] of the same group, and that does not "contain sufficient substance for publication".
I disagree with referee 3, and for many reasons.
First of all, referee 3 her/himself has already evidenced at least three points where the system studied in the present work is different from the system of Ref. [17]: it is evidently different. To this we must add that the scope of the present paper, the regime (low densities) to which it is in particular interested, and consequently the physical properties and characteristics to which it focuses, are different.
Of course, the concept of "follow-up paper" is arbitrary and can be strongly referee-dependent. But if we compare the couple of papers pointed by referee 3, with the thousands, or better, the tens-of-thousands papers [DOI: 10.2478/jdis-2018-0005] published in the recent years on the very same system, graphene, and studying all the same physical properties, also referee 3 would agree that, if a follow-up suspect has to be raised, this is more the case of the tons of papers on graphene! On this basis, referees had to systematically reject at least 99% of graphene papers, so to reduce the volume to an acceptable size, beyond any doubt of follow-up. Would have this been good for Science or not? Many think that yes, and I also hesitate. But the majority (which also has a large overlap with the authors of the tens-of-thousands graphene papers) is of different opinion. I would say that, in doubt, better to keep, let the author decide whether the paper is worth or not of publication.
Indeed (and here I come to the last reason why referee 3 is wrong) who did write the rule that follow-up papers must be rejected? Let's take the most famous case of a follow-up paper: the Einstein's Annalen der Physik 18, 639 (1905) $E = mc^2$ paper. Einstein himself confesses, and since the first sentence, that this is a follow-up paper of his previous Annalen der Physik 17, 891 (1905) paper, published only a couple of months before, in which he presented the theory of special relativity. We are very lucky that at Einstein's time there was no blind peer-review and papers were straightforwardly published by editors. A modern referee, like referee 3, would have certainly objected about the follow-up nature of the $E=mc^2$ Einstein paper, and rejected it, thus causing an enormous loss for humanity.

---

## Round 2 · Referee Report · Anonymous (Referee 4) · 2019-9-11

Report

Dear Editor

The paper in its revised form is now applicable for publications.

---

## Round 2 · Referee Report · Anonymous (Referee 3) · 2019-9-15

Report

With their revised version and the replies, the authors have clarified previous questions and improved their presentation.

Nevertheless, also thanks to these clarifications, it remains true that this is a simple work in the following sense: when exploiting rotational symmetry, as outlined in appendix A.2, the problem boils down to the diagonalization of an $n \times n$ matrix; according to the first paragraph of section III, the authors have worked with $n=128$. Hence, I do feel obliged to point out that there is an "exact diagonalization" community who works in dimensions that are orders of magnitude bigger than this. Indeed, if the matrix representation of $\hat{H}$ is sparse, as the authors say in their reply to item 6 of my previous report, one could use sparse-matrix algorithms such as the Lanczos procedure for diagonalization even before resorting to DMRG, as suggested at the end of section V ("Conclusion"). I have to admit that I am not sure if other items such as the overlap matrix of the local Gaussian basis would constitute a bottleneck. Still, even with a LAPACK library routine and when one exploits rotational symmetry one should be able to go to $n=1000 \ldots 10000$ basis functions and 2 electrons, or alternatively at least to 3 electrons for $n=128$.

Notwithstanding the above, simple computations can of course still be very useful if they lead to physical insights.

Requested changes

a) First paragraph of Introduction: Ref. 8 is from 2013 - is this really "recent" ?
b) Section IV.C, second paragraph: Is the "linear" increase really correct ? Note that the horizontal axis of Fig. 3 has a log scale. Thus, I would rather read off a logarithmic increase in $L$ (similar as the $\ln N$ increase stated in section II.D).
c) Section IV.C, last paragraph (item 8 of my previous report): The discussion of the cutoff by the basis set is still a bit unclear. Given that the dashed lines in Fig. 3 depend on the number of basis functions, I think that this should be stated clearly in the caption.

---

## Round 2 · Author Response

Dear Editor,

Thank you for sending us the reports of the referees.

We thank the referees for the constructive criticism and we are happy that two referees recommend publication of our work in SciPost. The third referee has not yet expressed an explicit recommendation.

Below we reply in detail to all the questions and comments of the referees. In particular, we have complied with the requested changes to the manuscript of all three referees. Therefore, we hope that the revised manuscript will be accepted for publication in SciPost.

Sincerely, Arjan Berger (on behalf of all the authors)

Authors’ answers to Reviewer 1.

  1. Reviewer’s comment

“I personally found the work really interesting (per se, as I said above) and stimulating further developments along the same line. For this reason, I don’t understand the authors’ need to add an introduction bringing on a ”technological interest” of the present study, making a link with ”quantum dots” and their ”tuning” to make them ”promising candidates for qubits”, up to even evoking the legendary and mythological ”quantum computer”. These introductions are in the style required by editors of those high-IF journals with divulging character. But on high-level journals, like SciPost, whose readership is composed by highly competent scientists who can perfectly understand the importance of the present work, such introductions are an offense to their intelligence. I think that they are unworthy and reduce the scientificity of the article.’

Authors’ response

We agree with the referee that mentioning the potential technological interest of our work is not necessary. We have removed this paragraph from the manuscript.

  1. Reviewer’s comment

“It is not immediately clear from the title, the abstract and even the introduction which is the system studied precisely. Indeed, until section II.A the reader is uncertain if the system is a really 1D ring with 1D kinetic and interaction terms. Or for quasi-1D the authors mean a 3D system but rigidly confined on a ring (in analogy with spherium with electrons rigidly confined on a 2D manifold but with a 3D interaction). I think that a more immediately comprehensible definition for the present system is ”a quasi-1D system composed of two electrons Gaussian-confined on a ring of perimeter L”. I think that this is an important information to be quickly delivered for a reader to promptly classify this work with respect to his interests.”

Authors’ response

We agree with the referee. In the revised manuscript we have made it clear from the outset, i.e., in the abstract and introduction, that we are dealing with ”a quasi-1D system composed of two electrons Gaussian-confined on a ring of perimeter L”.

The third and fourth line of the abstract now read:

"In particular, we study a quasi-one-dimensional system of two electrons that are confined to a ring by three-dimensional gaussians placed along the ring perimeter."

and when we outline the goals of our work in the introduction (right column) we now write

"We remove any border effects by confining the electrons to a ring by placing the three-dimensional gaussians along the ring perimeter"

  1. Reviewer’s comment

“I don’t know if the use of the term ”ab initio” in the title is appropriated in this case: ab initio normally refers to treatment of all (or a maximum relevant) of the degrees of freedom in a real system by referring to the microscopical (and not a model) Hamiltonian. The present is rather a model. Even Wigner, Nozieres et al., when studying the 3D electron-liquid jellium model, did not define it an ab initio study. And jellium has much more close realizations in nature (e.g. bulk sodium or lithium) than the present model. On the other hands, I agree that the present study ”is important to be able to describe Wigner localization within ab initio theory”.”

Authors’ response

We agree with the referee and we have removed ab initio from the title.

  1. Reviewer’s comment

“The present work casts some interesting doubts on the ab initio quantum chemistry (full-CI) Gaussian methodology, that is whether it possesses a real full 3D ab initio valence or it should be rather considered a reduced quasi-1D study. As it emerges here, a Gaussian basis-set, with Gaussians appropriately placed along a ring, in practice makes the system quasi-1D. Then also what in quantum chemistry are defined as ab initio full-CI studies of molecules, for examples in dimers with Gaussians placed at the two atomic positions or at best placing a further midbond set, might be rather receiving a strong bias toward a quasi-1D situation. And so, for the use of the Gaussian basis-set itself, they deviate from a genuine description of the real 3D system. It would be interesting if the authors could provide the comparison of their numerical study as a function of Gaussian width α, with the available analytical study on the rigid 1D ring: the case of one electron is already sufficient to show how much, depending on α, a Gaussian study is closer to the quasi-1D or rather evolves towards the full 3D “

Authors’ response

Depending on the problem at hand, i.e., the system and physical property under study, one chooses the most convenient basis set. For atoms and (small) molecules the most convenient basis is, in general, an atom-centered Gaussian basis set. This indeed gives a bias on the results, but, in general, we would expect it to be a very small bias. The problem discussed in this work is slightly different in this respect, since there are no nuclei. The positions of the centers of the Gaussians define the system, i.e., a ring. The test that the referee proposes is interesting but difficult to do in practice since evolving ”towards the full 3D” would mean increasing the width of the Gaussians by reducing α, thereby increasing the overlap between the Gaussians, which will lead to numerical problems due quasi-linear dependence.

  1. Reviewer’s comment

”It is to me evident that the diagonal of the 1-RDM (the density) is constant, and only the diagonal of the 2-RDM can be used to characterize a WC, but it is not clear why raising the dimensionality of the system (from the 1-torus/ring to the 2 and 3-torus) one needs higher order RDMs (3-RDM and 4-RDM). I would have said that the 2-RDM is sufficient always. Is this valid also for the full 2D and 3D electron liquids not confined on the torus? Can the authors provide an intuitive explanation, or at least a reference where this can be understood?”

Authors’ response

To explain this point we ask the referee to consider, as an example, the hexagonal 2D Wigner crystal. If we fix the position of an electron E in the 2-RDM, the nearest-neighbor electrons will have the highest probability density when they are positioned on the vertices of a hexagon with E at its center. However, there are an infinite number of hexagons with E at its center, all having equal probability. Therefore, the 2-RDM corresponding to an electron in E will be given by a set of circles, all having E as center, and each one having a constant value of the 2-RDM. For this reason, we need the 3-RDM, because fixing the position of a second electron, for example at one of the vertices of a hexagon around E, will fix the positions of the other electrons in that hexagon (and beyond). With a similar argument, we see that the 2-RDM is needed to characterize a 3-D crystal.

  1. Reviewer’s comment

”Since the beginning I had the impression that the particle-hole entropy is a quantity that strongly depends on the number M of natural orbitals taken into account, and so strongly depends on the basis-set size. At least much more than the two other considered indicators. If the authors confirm this impression of mine, than since the beginning it is evident the difficulty in using this quantity as a robust indicator of WC.“

Authors’ response

We agree with the referee that the particle-hole entropy depends on the basis-set size and that, therefore, one could expect it to be, a priori, not the most robust indicator to characterize Wigner localization. However, in a linear quasi-1D system we studied in a previous work [J. Chem. Phys.148, 124103 (2018)] it turned out to be a useful quantity to characterize the Wigner localization. Therefore, we included the study of the particle-hole entropy also in this work. In the revised manuscript we have added the following lines in section II.D to stress from the outset the dependence of the particle-hole entropy on the basis set, in particular in the limit of Wigner localization.

"The dependence of the particle-hole entropy on the natural spinorbitals implies a dependence on the basis set. For large densities this dependence is negligible but in the limit of Wigner localization there is a strong dependence on the basis set. This dependence can be made explicit as we will now show."

  1. Reviewer’s comment

”Finally, from this work it does not emerge a net criterion to exactly pinpoint the density at which the phase transition to the Wigner crystal really occurs. If one looks to the 2-RDM, for small L the FWHM is not well defined, so that the normalized FWHM is manually set to 1: therefore, we cannot take the WC phase transition to occur when the normalized FWHM departs from 1. The trace of the localization tensor does not achieve a well defined plateau for high densities, and so what to take for phase-transition boundary? I have the impression that the system is a Wigner crystal even at high densities and since the beginning, so that there is no real phase transition to another phase. Isn’t this really the case¿‘

Authors’ response

At high densities the system is a Fermi gas while it becomes a Wigner crystal at low densities. This can be seen from the diagonal of the 2-RDM in Fig. 1. It is close to a constant at high density, indicating that the two electrons behave independently while it is nonzero only around L/2 (the first electron is fixed at 0) at low density indicating that the two electrons are strongly correlated. Of course, we should probably speak of a Fermi-gas-like and a Wigner-crystal-like system, since Fermi gases and Wigner crystals pertain to many-electrons systems i.e., systems with 10^23 electrons. For the same reason it is difficult to pinpoint the phase transition which is well-defined only in the limit of an infinite number of electrons. Such a system is best represented within periodic boundary conditions. This is out of the scope of this work but it is something we will investigate in the future.

Authors’ answers to Reviewer 2

  1. Reviewer’s comment

”I would suggest to add a section or appendix about the details of numerical, for the pedagogical purpose.”

Authors’ response

We have added an appendix in which we show the details of the numerical procedure.

  1. Reviewer’s comment

”I think that the authors should discuss more about transition point!!!, whether it can be found with the introduced observables or not?”

Authors’ response

The transition point is well-defined only in the limit of an infinite number of electrons. Such a system is best represented within periodic boundary condtions. This is out of the scope of this work but it is something we will investigate in the future. Nevertheless, the current investigation demonstrates that the localization tensor is the preferred observable to study the transition point in such a system. To clarify this point we have added the following sentence at the end of the Conclusion,

"This would allow us to pinpoint the phase transition which is well-defined only in the thermodynamic limit."

Authors’ answers to Reviewer 3

  1. Reviewer’s comment

”This is actually a pretty simple work, following up on a previous investigation [17] by the same group.“

Authors’ response

Indeed, as written in the manuscript, this work follows up on a previous investigation of ours. Although the work might be considered simple because it only deals with 2 electrons, it is not that simple to accurately describe such a system at extremely low densities where strong correlation plays a crucial role. This is also the main difference with the previous work, where we were restricted to much larger densities.

  1. Reviewer’s comment

”The authors study the behavior of two electrons that are implicitly confined to a ring of varying diameter by the choice of computational basis. The main differences with respect to the previous work Ref. [17] from the same group are: ∗ Ring geometry instead of open line segment. ∗ No positive background charge. ∗ Refined spatial resolution thanks to a customized code. ∗ 2 instead of up to 8 electrons.“

Authors’ response

Indeed, this is a good summary of the differences with our previous work, with the crucial difference being point 3 (Refined spatial resolution thanks to a customized code), since it allows us to study electron correlation at extremely low densities.

  1. Reviewer’s comment

”The authors claim that this study allows them to detect signatures of a Wigner crystallization. However, for a true Wigner crystal, one would need a many-electron system while the present work considers only the relatively simple two-particle problem.“

Authors’ response

We agree with the referee that to detect signatures of Wigner crystallization one would need a many-electron system. For this reason, when we address our results we speak of Wigner localization, i.e., localization that is due to the electronic repulsion. Therefore, we do not claim that we can detect signatures of a Wigner crystallization but we claim that we can detect signatures of Wigner localization. However, since the origin of Wigner localization and Wigner crystallization is the same we expect that the quantities with which we can detect Wigner localization will also be good candidates to detect Wigner crystallization. This is to be confirmed in future work.

  1. Reviewer’s comment

”Given that this is a follow-up study to Ref. [17] and that the problem investigated is a simple two-particle problem, I am not sure if this manuscript contains sufficient substance for publication in SciPost Physics.“

Authors’ response

As mentioned before, not all two-particle problems are simple. In this case the difficulty lies in the challenge to accurately describe the strong correlation at extremely low densities with numerical tools.

  1. Reviewer’s comment

”In any case, the authors should choose a title that makes the two-particle problem explicit. The terminology ”Wigner molecule” (see end of the second paragraph of the Introduction) might be an option.“

Authors’ response

We thank the referee for the suggestion which is indeed appropriate. We changed the title to ”A Wigner molecule at extremely low densities: a numerically exact study”

  1. Reviewer’s comment

”Apart from this, chapter III is a bit short, in particular for a work whose main merit is supposed to be a specially written dedicated code. Specifically, I would find the following information useful: * Was the rotational symmetry of the ring exploited? In other words: conservation of angular momentum should allow to reduce the two-electron problem to an effective one-body problem. * What is the structure of the matrix representation of the Hamiltonian Eq. (2) represented in the basis Eq. (1)? Specifically: is it a sparse matrix or can it be approximated by a sparse matrix to numerical accuracy? * What algorithm was used to find the ground state? A library routine or an iterative algorithm?”

Authors’ response

We agree with the referee that it is useful to discuss the details of our numerical procedure.

Yes, we exploited the rotational symmetry but we also did calculations without symmetry using the local gaussian functions. The results of the two approaches are the same, as they should. We have included this information in the last paragraph of chapter III and we added the details in an appendix.

The Hamiltonian expressed in the basis of gaussian functions is sparse. Since numerically the problem at hand is sufficiently simple, we have not done any approximations, for example by neglecting small contributions due to gaussians with little overlap. However, in future work on larger systems, both in dimension and in the number of electrons, we could benefit from the sparsity of the Hamiltonian matrix.

To find the ground state we first performed a spin adaptation of the wave function. In particular this is important at low density, where the singlet and triplet wave functions tend to become degenerate. In absence of spin adaptation, due to numerical errors, the diagonalization procedure could mix the two quasi-degenerate states. The Hamiltonian matrix reduces therefore to two spin adapted diagonal blocks which we diagonalized with the LAPACK library routine DSYEV which computes the eigenvalues and eigenvectors of real symmetric matrices. The ground state can then be easily found as the state with lowest energy. We have also included this information in the last paragraph of chapter III and we added the details in the appendix.

  1. Reviewer’s comment

”The references [17,38-46] on applications of the ”localization tensor” are all from the group around Stefano Evangelisti. If this tool was also used by other groups, appropriate references should be added.”

Authors’ response

We have added references to two recent works of Penda ́s, Francisco and co-workers (Refs. [40] and [41] of the revised manuscript), which were the only works we found with practical applications of the localization tensor. We have also added two references to works which discuss general theory of the localization tensor (Refs. [27] and [28] of the revised manuscript).

  1. Reviewer’s comment

”I did not find a definition of ”natural spinorbitals” such that the particle-hole entropy Eq. (11) is ill defined. A related comment concerns the particle part of the entropy in Fig. 3. It seems to me that the asymptotic value (2ln(128) ?) measures just the number of basis functions and thus is not intrinsic to the problem. If so, the asymptotic value shown in Fig. 3 is actually a bit misleading.

Authors’ response

We have added the definition of ”natural spinorbitals” just below Eq. (11), i.e.,

"The natural spinorbitals are the eigenfunctions of the one-body reduced spin-density matrix and the occupations numbers are its eigenvalues."

Indeed, the entropy depends strongly on the number of basis functions in the asymptotic limit. We have added this information to section II. D where we now write

"The dependence of the particle-hole entropy on the natural spinorbitals implies a dependence on the basis set. For large densities this dependence is negligible but in the limit of Wigner localization there is a strong dependence on the basis set. This dependence can be made explicit as we will now show."

  1. Reviewer’s comment

”The units of the full-width at half maximum in Fig. 1 are not clear (I suspect that it is normalized by the length L).

Authors’ response

Indeed, as written in the caption of Fig. 1, the full-width at half maximum normalized by the length. Therefore it has no units. In the revised manuscript we have made the normalization with respect to L explicit on the y-axis of the inset in Fig. 2.

  1. Reviewer’s comment

”It is good to have a perspective on two dimensions in the Conclusions. Among others, this justifies some of the more general discussions throughout the manuscript. However, it would also be good to have the authors’ opinion on the possibility to go beyond two particles to the true many-body Wigner crystal.

Authors’ response

We agree with the referee that it is of interest to discuss the possibility to go beyond two electrons, towards the true many-body Wigner crystal. In fact, when we mention the two-dimensional case we implicitly had in mind ”a two-dimensional system with more than two electrons”. We have now made this point explicit in the conclusions, i.e.,

"In particular, it would be interesting to study two-dimensional electron systems with more than two electrons, going towards a true many-electron Wigner crystal that can be experimentally observed."

---

## Round 2 · List of Changes

1. We changed the title to ”A Wigner molecule at extremely low densities: a numerically exact study”
  2. The third and fourth line of the abstract now read "In particular, we study a quasi-one-dimensional system of two electrons that are confined to a ring by three-dimensional gaussians placed along the ring perimeter."
  3. We have removed the first paragraph of the Introduction.
  4. when we outline the goals of our work in the introduction (right column) we now write "We remove any border effects by confining the electrons to a ring by placing the three-dimensional gaussians along the ring perimeter"
  5. We have added the following lines in section II.D: "The dependence of the particle-hole entropy on the natural spinorbitals implies a dependence on the basis set. For large densities this dependence is negligible but in the limit of Wigner localization there is a strong dependence on the basis set. This dependence can be made explicit as we will now show."
  6. We have expanded chapter III on computation details and added an appendix in which we show the details of the numerical procedure.
  7. We have added the definition of ”natural spinorbitals” just below Eq. (11), i.e., "The natural spinorbitals are the eigenfunctions of the one-body reduced spin-density matrix and the occupations numbers are its eigenvalues."
  8. We have added a couple of sentences in section II. D where we now write "The dependence of the particle-hole entropy on the natural spinorbitals implies a dependence on the basis set. For large densities this dependence is negligible but in the limit of Wigner localization there is a strong dependence on the basis set. This dependence can be made explicit as we will now show."
  9. We have made the normalization of the FWHM with respect to L explicit on the y-axis of the inset in Fig. 2.
  10. We have added the following sentence to the Conclusions: "In particular, it would be interesting to study two-dimensional electron systems with more than two electrons, going towards a true many-electron Wigner crystal that can be experimentally observed."
  11. We have added the following sentence to the Conclusion: "This would allow us to pinpoint the phase transition which is well-defined only in the thermodynamic limit."
  12. We have added the following sentence to the conclusion. "Finally, it could be of interest to combine our approach with the density-matrix renormalization group (DMRG) algorithm to treat 1D systems with many electrons. [47,48]"
  13. We have added a footnote (Ref. [46]) to give more details about the 2-RDM that is reported in Fig. 1. This is important if somebody would like to reproduce our calculations. The footnote reads "To be precise, the results in Fig. 1 are the 2-RDM expressed in the basis of the orthonormal gaussians, i.e., Γ_{iαjβiαjβ} = ⟨Ψ|a†{iα} a†|Ψ⟩ in which the first electron is fixed at i = 1."} a_{iα} a_{jβ
  14. We have added references to two recent works of Penda ́s, Francisco and co-workers (Refs. [40] and [41] of the revised manuscript), which were the only works we found with practical applications of the localization tensor. We have also added two references to works which discuss general theory of the localization tensor (Refs. [27] and [28] of the revised manuscript).

---

## Editorial Decision

published